Dynamic thresholding search for the feedback vertex set problem

Sun Wen 1
http://orcid.org/0000-0001-8813-4377 Hao Jin-Kao 2 jin-kao.hao@univ-angers.fr
Wu Zihao 1
Li Wenlong 1
Wu Qinghua 3
1 School of Cyber Science and Engineering, Southeast University , Nanjing , China
2 LERIA, Université d’Angers , Angers , France
3 School of Management, Huazhong University of Science and Technology , Wuhan , China
Stützle Thomas
Electronic publication date: 2023 Feb 10
Publication date: 2023
Volume: 9
Electronic Location ID: e1245
Received 2022 Sep 26; Accepted 2023 Jan 17
Copyright: © 2023 Sun et al.
Copyright year: 2023
Copyright holder: Sun et al.
License: This is an open access article distributed under the terms of the Creative Commons Attribution License, which permits unrestricted use, distribution, reproduction and adaptation in any medium and for any purpose provided that it is properly attributed. For attribution, the original author(s), title, publication source (PeerJ Computer Science) and either DOI or URL of the article must be cited.
License URL: https://creativecommons.org/licenses/by/4.0/

Keywords: Feedback vertex set, Dynamic thresholding search, Descent search, Heuristic

Funding: National Natural Science Foundation of China 62101125, 72122006 Natural Science Foundation of Jiangsu Province SBK2020040023 Fundamental Research Funds for the Central Universities 2242022R40067, 2242022K30007 This work was supported by the National Natural Science Foundation of China (Grant No. 62101125, No. 72122006), the Natural Science Foundation of Jiangsu Province (Grant No. SBK2020040023), and the Fundamental Research Funds for the Central Universities (Grant Nos. 2242022R40067, 2242022K30007). The funders had no role in study design, data collection and analysis, decision to publish, or preparation of the manuscript.

==============================
Given a directed graph G = (V, E), a feedback vertex set is a vertex subset C whose removal makes the graph G acyclic. The feedback vertex set problem is to find the subset C* whose cardinality is the minimum. As a general model, this problem has a variety of applications. However, the problem is known to be NP-hard, and thus computationally challenging. To solve this difficult problem, this article develops an iterated dynamic thresholding search algorithm, which features a combination of local optimization, dynamic thresholding search, and perturbation. Computational experiments on 101 benchmark graphs from various sources demonstrate the advantage of the algorithm compared with the state-of-the-art algorithms, by reporting record-breaking best solutions for 24 graphs, equally best results for 75 graphs, and worse best results for only two graphs. We also study how the key components of the algorithm affect its performance of the algorithm.

Introduction

Given a directed graph G=(V,E), where V denotes the set of vertices and E the set of edges, a feedback vertex set (FVS) is a vertex subset C⊂V whose removal leads to an acyclic graph. The feedback vertex set problem (FVSP) aims to identify a FVS of minimum cardinality. In other words, we want to remove the fewest vertices to make the graph acyclic.

The decision version of the FVSP is one of the 21 nondeterministic polynomial-time complete (NP-complete) problems, which were first proved in the early 1970s (Cook, 1971; Karp, 1972). Its broad applications include very large scale integration circuit design (Festa, Pardalos & Resende, 1999), deadlock detection (Leung & Lai, 1979; Wang, Lloyd & Soffa, 1985), program verification (Seymour, 1995), Bayesian inference (Bar-Yehuda et al., 1998), operating systems (Silberschatz, Galvin & Gagne, 2006) and complex network systems (Liu, Slotine & Barabási, 2011). A typical application of the FVSP is to control the state of a complex network system, making the system to change from any given state to an expected state by controlling a minimal subset of vertices from the outside. For instance, Mochizuki et al. (2013), Fiedler et al. (2013) and Zhao et al. (2020) investigated FVS-based control mechanisms. This FVS approach proves to be suitable when only the network formation is known, while the functional form of the governing dynamic equations are ambiguous (Zhao et al., 2020). Studies also showed that this approach needs to remove fewer vertices than other structure-based methods in many cases (e.g., Zañudo, Yang & Albert, 2017).

Figure 1A shows a directed graph G with five vertices {a,b,c,d,e}. Figure 1B displays an arbitrary FVS {a,b,d} with a cardinality of 3, while Fig. 1C presents an optimal FVS {b,c} with a minimum cardinality of 2.

Figure 1 Random FVS and optimal FVS (the vertices of FVS are in orange and the other vertices are in blue).

(A) A given graph G; (B) a random FVS {a,b,d}; (C) an optimal FVS {b,c}.

Some approximation algorithms were proposed for the FVSP to provide solutions of provable quality. Erdős & Pósa (1962) presented an algorithm with an approximation ratio of 2logn ( n=|V|). Later, Monien & Schulz (1981) improved the approximation ratio to logn. Even et al. (1998) realized an approximation factor of O(logτloglogτ) on directed graphs, where τ is the size of a minimum FVS for the input graph. Other polynomial time approximation algorithms for the FVSP in tournament graphs include those presented by Cai, Deng & Zang (2001), Mnich, Williams & Végh (2015) and Lokshtanov et al. (2021).

From the perspective of solution methods for the FVSP applied to the ISCAS89 benchmark instances (up to 1,728 vertices), several exact algorithms combined with graph reduction have been proposed. Specifically, Levy & Low (1988) presented an exact reduction based on the graph structure and proved its equivalence to the original graph. Based on the exact reduction of Levy & Low (1988), Orenstein, Kohavi & Pomeranz (1995) proposed graph partitioning methods with new reduction operations, which achieved optimal results on all the ISCAS89 benchmark instances within 2 CPU hours on a Sun-4 station. Lin & Jou (1999) investigated the branch-and-bound algorithm that considers the exact reduction of Orenstein, Kohavi & Pomeranz (1995), which could find the optimal results for the ISCAS89 benchmarks in less than 3 s on a SUN-UltraII workstation. There are many vertices whose in-degrees or out-degrees are 0 or 1 in the ISCAS89 benchmark instances. The average reduction ratio (the sum of the deleted vertices/the sum of vertices of a given graph) of these reduction approaches is 72.48%, implying that these benchmark instances are easy for modern FVSP algorithms. Hence, we report in this work computational results not only on these ISCAS89 instances, but also on more challenging benchmark instances. Some theoretical exact algorithms were reported without experimental validation. For example, Razgon presented a backtrack algorithm that solved the FVSP in time O(1.8899n) (Razgon, 2006) and a branch-and-prune algorithm requiring O(1.9977n) time (Razgon, 2007). Fomin, Gaspers & Pyatkin (2006) developed a branching algorithm with a time complexity of O(1.7548n). Some exact algorithms were also proposed specifically for tournament graphs (Moon, 1971; Gaspers & Mnich, 2013; Fomin et al., 2019).

Parameterized algorithm is another popular approach to solve the FVSP. Chen et al. (2008) solved the FVSP in time O(4kk!nm), where m is the number of edges of the graph, and k is the largest number of selected vertices. Lokshtanov, Ramanujan & Saurabh (2018) proposed an algorithm whose running time is O(4kk!k5(n+m)). Bonamy et al. (2018) proved that the running time could be reduced to 2O(k)nO(1) for planar directed graphs.

Considering the high difficulty of the FVSP, exponential time exact algorithms can only be applied to specific graphs (such as the ISCAS89 benchmark instances). Therefore, heuristic algorithms are usually adopted to obtain sub-optimal solutions within a reasonable time frame to tackle more general and complicated graphs. Pardalos, Qian & Resende (1998) introduced the first heuristic algorithm for the FVSP, based on the greedy randomized adaptive search procedure (GRASP) approach. Galinier, Lemamou & Bouzidi (2013) presented a simulated annealing (SA) that applies the INSERT operation and a fast neighborhood evaluation technique. Besides, they combined their SA algorithm with an exact reduction technique (Red+SA). They performed extensive experiments on 40 random instances (1,000 runs for SA and 30 runs for Red+SA) and showed that both SA and Red+SA fully dominate the GRASP algorithm. Zhou (2016) constructed a spin glass model and implemented a belief propagation-guided decimation (BPD) algorithm, which initially declares all vertices as active and repeats the fixing-and-updating procedure until all vertices are turned into inactive. The BPD algorithm obtained the same best known results as SA on the set of instances in Galinier, Lemamou & Bouzidi (2013). Tang, Feng & Zhong (2017) put forward a variant of SA by introducing a non-uniform neighborhood sampling strategy (SA-FVSP-NNS). However, experiments proved that SA in Galinier, Lemamou & Bouzidi (2013) dominates the SA-FVSP-NNS for 30 runs with 39 better results and one equal result. In summary, according to the literature (Galinier, Lemamou & Bouzidi, 2013; Zhou, 2016; Tang, Feng & Zhong, 2017), among all existing practical algorithms for the FVSP, the Red+SA, SA (Galinier, Lemamou & Bouzidi, 2013) and BPD (Zhou, 2016) algorithms are the top algorithms for the problem. Thus, we use them as our main reference approaches for this study. Finally, Qin & Zhou (2014) presented the simulated annealing local search algoritm (SALS), which is an adaptation of the SA algorithm of Galinier, Lemamou & Bouzidi (2013) to the undirected FVS problem and showed its effectiveness on large random undirected graphs. We adopt SALS as an additional reference method for our study on undirected graphs.

The above literature review demonstrates that progresses were continually realized since the introduction of the FVSP. However, few effective heuristic algorithms exist, able to solve the problem in a satisfactory manner. This work partially fills the gap by presenting an iterated dynamic thresholding search (IDTS) algorithm for the FVSP. The algorithm includes three exact reduction rules to simplify the graph, a greedy initialization to generate initial acyclic subgraphs, a dynamic thresholding local search to reduce the size of the acyclic subgraph, and a learning-based perturbation to reconsider the vertices that would have been wrongly regarded as feedback vertices.

Experiments are performed on 101 benchmark instances from various sources to assess the IDTS algorithm. For the 70 instances with unknown optima, IDTS is able to improve 24 best-known solutions and attain the best-known results for 44 other instances. Only for two instances, IDTS reports a worse result. Moreover, IDTS easily attains the known optimal results for all 31 ISCAS89 benchmark instances.

The remainder of this article is arranged as follows. “Basic Notations and Fitness Function” introduces useful basic notations and fitness function of the FVSP. “Preliminaries” is a preliminary presentation. “Iterated Dynamic Thresholding Algorithm for the FVSP” explains the components of the IDTS algorithm. “Experimental Results and Comparisons” evaluates the algorithm with computational results. “Analysis” studies critical components of the proposed algorithm, and “Conclusions” provides conclusions.

Basic notations and fitness function

This section introduces relevant basic definitions, solution representation and fitness function, which are necessary for presenting the proposed algorithm.

Basic definitions

Given a directed graph G=(V,E), basic definitions that are useful for describing the proposed IDTS algorithm are presented as below.

Definition 1: a critical vertex of G is a vertex that belongs to a FVS. We use C to denote the set of critical vertices that have been detected. C is a FVS only when all vertices of the FVS are detected.

Definition 2: an uncritical vertex is a vertex that does not belong to a FVS. We use U to denote the set of uncritical vertices, and V=C∪U,C∩U=∅.

Definition 3: a redundant vertex refers to a vertex that is recognized as critical or uncritical according to the exact rules proposed by Levy & Low (1988). We use Vr to denote the set of redundant vertices that have been detected, Cr to denote the set of critical vertices of Vr, Ur to denote the set of uncritical vertices of Vr, and Vr=Cr∪Ur,Cr∩Ur=∅.

Definition 4: V0 refers to the set of residual vertices after applying the removal exact algorithm proposed by Levy & Low (1988). C0 denotes the set of feedback vertices of V0 (that is, all vertices of a FVS are detected and belong to C0), U0 denotes the set of non-feedback vertices of V0, and V0=C0∪U0,C0∩U0=∅. Levy & Low (1988) proved that the FVS of the reduced graph plus the FVS removed in the reduction process composes the FVS of the original graph. Let Cr be the set of vertices removed in the reduction process, and C0∗ be the minimum FVS of the reduced graph G=(V0,E0), where E0=V0×V0∩E. Then, Cr∩C0∗=∅ and C0∗∪Cr is a minimum FVS of the given graph G=(V,E). In this case, only the feedback vertices of the reduced graph need to be found out.

In summary, the vertex set V of G consists of two disjoint sets {V0,Vr} or four disjoint sets {C0,U0,Cr,Ur}.

Definition 5: a directed acyclic graph (DAG) (Bangjensen & Gutin, 2008) is a directed graph with no directed cycles.

Vertices in each DAG are in a topological ordering where the starting point of every directed edge is ahead of its terminal point (Galinier, Lemamou & Bouzidi, 2013). For a vertex set U⊂V, we notice that the induced subgraph GU=(U,EU), EU=U×U∩E is acyclic if and only if V∖U is a FVS. Hence, the objective of the FVSP is to find the set U that has the maximum cardinality to make GU=(U,EU) acyclic.

Figure 2 presents an example illustrating these basic definitions. For the given graph G=(V,E), let {a,b,d,h} be the current FVS. The set of critical vertices C is the current FVS {a,b,d,h}, and the set of remaining vertices is the set of uncritical vertices U, i.e., {c,e,f,g,i,j}. According to the rules in “Reduction Procedure”, h can be recognized as a critical vertex ( Cr={h}, purple vertex) and f,g,i,j as uncritical vertices ( Ur={f,g,i,j}, dark blue vertices). Thus the set of redundant vertices Vr is {f,g,h,i,j}, and the set of residual vertices V0 is {a,b,c,d,e}. V0 can be divided into C0={a,b,d} (orange vertices) and U0={c,e} (blue vertices). Clearly, the graph induced by the vertices in U={c,e,f,g,i,j} is a DAG without directed cycles.

Figure 2 An example for illustrating basic definitions.

Solution representation and fitness function

The solution representation and fitness function of the FVSP are given as follows.

Solution representation: the constraint of the FVSP is that there is no cycle in U0 after removing the set of redundant vertices Cr∪Ur and the set of critical vertices C0. To quickly assess the number of cycles in U0 after each neighborhood operation, the number of conflicts (see “Preliminaries”) is taken as the number of cycles (Galinier, Lemamou & Bouzidi, 2013). Let π be an assignment of the vertices of U0 to the positions {1,2,…,|U0|}, and thus the permutation π denotes the candidate solution (Galinier, Lemamou & Bouzidi, 2013).

Fitness function: To evaluate the quality of the FVS C, the evaluation or fitness function counts the number of vertices in C. Recall that Ur is the set of uncritical vertices of the redundant vertices and π is the corresponding permutation solution of C. The fitness function f0 (to be minimized) is given by

(1) Minimum f0(π)=|V|−|Ur|−|π|

Thus, the minimization of the function f0 is equal to the maximization of the fitness function f, which is expressed as:

(2) Maximize f(π)=|π|

Preliminaries

In this section, we introduce two properties of FVS: number of conflicts and INSERT operator position.

Number of conflicts: u and v are a pair of conflicting vertices if v is ahead of u in permutation π and there is a directed edge from u to v. The number of conflicting vertex pairs of the permutation π is the number of conflicts. Let edge(u,v)=1 if there is a directed edge from u to v. Otherwise, edge(u,v)=0. We use c(u,v), where u,v∈π,u≠v,edge(u,v)=1, to indicate whether u and v form a conflicting vertex pair as follows

(3) c(u,v)={1,πv<πu0,πv>πu

where πv represents the position chosen for vertex v in permutation π. Then, the number of conflicts g(π) is given by

(4) g(π)=∑u,v∈πc(u,v)

Thus, for a conflict-free solution π, g(π)=0 holds. The time complexity to compute g(π) is O(dmax), where dmax denotes the largest degree of a vertex in the graph. Clearly, the number of conflicts is more than the actual number of cycles for the same solution.

Figure 3 displays two cases between the number of conflicts and the number of cycles. The left part of Fig. 3A indicates the current remaining vertex set U0 of the given G, and the right part is its corresponding permutation π={a,b,d}. In this permutation, no directed edge from b to a ( d to b or to a) exists. Thus, there is no conflicting pair, meaning the number of cycles is 0. Figure 3B shows a situation where the number of conflicts is more than the number of cycles. Vertex b is behind vertex d in the solution permutation π={d,a,b} on the right side, and there is a directed edge from b to d. Accordingly, vertex b and vertex d are conflicting, and the number of conflicts is 1, which exceeds the number of cycles (0).

Figure 3 An example of conflicts in a solution.

(A) The number of conflicts is equal to the number of cycles; (B) the number of conflicts exceeds the number of cycles.

INSERT operator position: The INSERT operator inserts a vertex v from the uncritical vertex set to two possible positions in π (Galinier, Lemamou & Bouzidi, 2013); one is closely behind its numbered in-coming neighbors (named i−(v)), and the other is just ahead its numbered out-going neighbors (named i+(v)). The number of conflicting pairs after the insertion operation at the above two positions is calculated respectively, and the position with less conflicting pairs is selected.

Algorithm 1: IDTS algorithm for the FVSP

  Input: A directed graph G=(V,E), cutoff time, and search depth ω	
  Output: The smallest FVS of size |C∗|	
1  π←∅,π⋆←∅, |C∗|←|V|	
2  (G,Cr,Ur)←Reduction_phase(G)          /*Section 3.2*/	
3  π←Greedy_Initialization(G)          /*Section 3.3*/	
4  π⋆←π	
5 while The cutoff time is not reached do	
6    π←Local_search(G,π,π⋆,ω)           /*Section 3.4*/	
7    π←Perturbation(π)          /*Section 3.5*/	
8   if f(π)>f(π⋆) then	
9     π⋆=π	
10   end	
11 end	
12  C∗←Recovery_phase(G,Cr,Ur,π⋆)         /*Section 3.6*/	
13 return |C∗|	

We use <v,i−(v),i+(v)> to represent such a move, and π⊕<v,i−(v),i+(v)> to stand for the neighboring solution generated by applying the INSERT move to π. Moreover, g(π⊕<v,i−(v),i+(v)>) refers to the number of conflicts after inserting the vertex v∈C0. NI(π) contains the vertices that satisfy the condition g(π⊕<v,i−(v),i+(v)>)=0. That is, the vertex v to be inserted has to be a vertex that will not cause any conflict after being inserted into π. NI(π) can be expressed as

(5) NI(π)={v:g(π⊕<v,i−(v),i+(v)>)=0,v∈C0}

Iterated dynamic thresholding algorithm for the fvsp

Basic steps

This section introduces the iterated dynamic thresholding algorithm for solving the FVSP, which is composed of five main procedures as shown in Algorithm 1.

Reduction procedure: IDTS adopts a set of conventional reduction rules (Levy & Low, 1988) to simplify the given graph G. Firstly, a set of redundant vertices Vr (made up of the set of critical vertices Cr and the set of uncritical vertices Ur) is confirmed according to those rules. Then, the redundant vertices and related edges (whose starting or ending vertex is a redundant vertex) are deleted, reducing the input graph G=(V,E) to the reduced graph G=(V0,E0) (see “Reduction Procedure”).

Initialization procedure: this procedure greedily chooses a vertex (Cai, Huang & Jian, 2006) such that its insertion to π does not increase the number of cycles. This process continues until no such vertex can be inserted (see “Greedy Initialization”).

Local search procedure: this procedure consists of two complementary search stages: a dynamic thresholding search stage (diversification) to extend the search to unexplored regions, and a descent search stage (intensification) to find new local optimal solutions with improved quality. These two stages alternate until the best-found solution cannot be further improved for ω continuous local search rounds (see “Local Search”).

Perturbation procedure: When the search is considered as trapped in a deep local optimum, the perturbation procedure is initiated to move some specifically identified vertices between U0 and C0 to relieve the search from the trap. The solution perturbed is then adopted to start the next round of the local search procedure (see “Perturbation Procedure”).

Recovery procedure: If the best solution ever found cannot be improved after γ continuous local search rounds and the perturbation phase, the search then terminates and the recovery procedure starts. The best solution (minimum feedback vertex set) found in the search procedure is recorded as C0∗ and returned as the input of the recovery procedure. The current reduced graph G=(V0,E0) is recovered to the original graph G=(V,E), and the FVS C0∗ for G=(V0,E0) is correspondingly projected back to C0∗∪Cr for G=(V,E) (see “Recovery Procedure”).

Reduction procedure

The graph reduction procedure follows three rules when traversing all vertices in a given graph G=(V,E) and processes those that satisfy any rule proposed by Levy & Low (1988). The three reduction rules are as follows.

Rule 1: If the in-degree (out-degree) of a vertex v is 0, that is, v is an uncritical vertex, then v and all its edges can be deleted without missing any optimal feedback vertex of G. Such vertices are added into the redundant uncritical set Ur ( Ur=Ur∪{v}). For example, as shown in Fig. 4B, the edges of the vertex g whose in-degree is 0, and those of the vertex j whose out-degree is 0 can be deleted.

Figure 4 An example of reduction operation for the given graph G.

(A) The given graph G; (B) applying Rule 1 to delete g (in-degree = 0) and j (out-degree = 0); (C) applying Rule 2 to delete f (in-degree = 1) and i (out-degree = 1); (D) Applying Rule 3 to delete h (self-loop).

Rule 2: If the in-degree (out-degree) of a vertex v is 1, and there is no self-loop, then vertex v can be merged with the unique precursor (successor) vertex without missing any optimal feedback vertex of G. The merging process is that all edges connected to the vertex v are linked to the unique precursor (successor) vertex of v. Such vertices are added into the redundant uncritical set Ur ( Ur=Ur∪{v}). Figure 4C displays that the vertex f with an in-degree of 1 can be merged with the precursor vertex a, and the vertex i with an out-degree of 1 can be merged with the successor vertex b.

Rule 3: If a self-loop exists for a vertex v, then v and all its edges can be deleted and recovered as a part of the feedback vertex set without losing any optimal feedback vertex of G. Such vertices are added into the redundant uncritical set Cr ( Cr=Cr∪{v}). As shown in Fig. 4D, the self-loop vertex h and its connected edges can be deleted.

After deleting the sets Ur and Cr, the remaining vertex set V0=V∖(Ur∪Cr), and the reduced subgraph G=(V0,E0), (E0=(V0×V0∩E)). After obtaining reduced G, the greedy initialization is used to generate an initial solution for it.

Greedy initialization

Given the reduced subgraph G=(V0,E0), its critical vertex set is defined as C0, and the uncritical vertex set as U0. Recall that π is an assignment of the vertices of U0 to the positions {1,2,…,|U0|}. We initialize C0=V0, π=∅. Then, π is iteratively extended in the greedy procedure by inserting a minimum-score vertex v of C0 until no vertex can be inserted.

(1) Calculate NI(π): recall that NI(π) contains the vertices that satisfy g(π⊕<v,i−(v),i+(v)>)=0. Thus, we only has to traverse all vertices and add the vertex whose g(π⊕<v,i−(v),i+(v)>)=0 into NI(π).

(2) Select one vertex: choose a vertex v with the minimum score (Eq. (6), Cai, Huang & Jian, 2006) in NI(π) and insert it into the current permutation π←π ⊕<v,i−(v),i+(v)>.

(6) score(v)=|deg−(v)+deg+(v)|−λ×|deg−(v)−deg+(v)|

where deg−(v) and deg+(v) are the in-degree and the out-degree of v respectively, and λ is a parameter ( λ=0.3 according to Cai, Huang & Jian (2006)).

(3) Update NI(π): after inserting the vertex v into π, we only have to recalculate the number of conflicts g(π⊕<u,i−(u),i+(u)>) of its neighbors u∈NI(π) according to Eq. (4). Any vertex satisfying g(π⊕<u,i−(u),i+(u)>)≠0 will be eliminated from NI(π). Thus, the complexity of this updating step is O(dmax2), where dmax is the largest degree of a vertex in the graph. This process continues until no vertex can be inserted into π, i.e., NI(π)=∅.

In this initialization, a legal conflict-free π with a certain quality can be obtained, and further improved in the dynamic thresholding search stage of the algorithm.

To explain this process, we consider the reduced graph G=(V0,E0) in Fig. 4D (with vertices {a,b,c,d,e}). For Fig. 4D, C0={a,b,c,d,e}, U0=π=∅ and g(π⊕<v,i−(v),i+(v)>)=0, ∀v∈C0. Figure 5 shows how the greedy procedure works. Firstly we calculate NI(π)={a,b,c,d,e}. Then, it is detected that score(a)=4.7,score(b)=4,score(c)=4,score(d)=4.7,score(e)=4 according to Eq. (6). Finally, we select a vertex from NI(π) with the minimum score and insert it into π. Suppose we select vertex e for insertion. NI(π) is updated to {a,c,d}. As shown in Fig. 5A, the solution after the first greedy insertion is the permutation π={e}. By repeating the above steps until NI(π)=∅, we obtain the local optimal permutation π={e,c} (Fig. 5B). Meanwhile, this process may unfortunately misclassify critical vertices in permutation π. For example, the optimal permutation of Fig. 4D is {d,e,a} with the set of critical vertices {b,c}, while the local optimal permutation of Fig. 5B is {e,c} with the set of critical vertices {a,b,d}.

Figure 5 An example of greedy initialization.

(A) The solution π={e} and the set of critical vertices C0={a,b,c,d} after the first greedy insertion; (B) local optimal solution π={e,c} and the set of critical vertices C0={a,b,d}.

Local search

The local optimization aims to improve the initial permutation provided by the greedy initialization, and it consists of two stages. The first stage (dynamic thresholding search) brings diversity as it accepts equivalent or worse solutions (line 4 of Algorithm 3), and the second stage applies a descent search that accepts only better solutions (line 5 of Algorithm 3) to guarantee a concentrated and directed search. These two stages alternate until the best found solution cannot be further improved for ω local search rounds.

The dynamic thresholding search stage

There are many successful applications of dynamic thresholding search (Dueck & Scheuer, 1990; Moscato & Fontanari, 1990) (e.g., the frequency assignment (Diane & Nelson, 1996), quadratic multiple knapsack problem (Chen & Hao, 2015), heterogeneous fixed fleet vehicle routing problem (Tarantilis, Kiranoudis & Vassiliadis, 2004), and others (Chen & Hao, 2019; Lai et al., 2022; Zhou, Hao & Wu, 2021)).

Algorithm 2: Greedy initialization for FVSP

  Input: A reduced graph G=(V0,E0)	
  Output: A solution permutation π	
 1  π←∅	
 2  NI(π)←∅	
 3 for each v∈V0 do	
 4   Calculate NI(π) according to Eq. (5)      /*Step 1*/	
 5 end	
 6 while NI(π)≠∅ do	
 7   Choose a vertex v∈NI(π) with the minimum score according to Eq. (6) and insert it into π      /*Step 2*/	
 8   Update NI(π)     /*Step 3*/	
 9 end	
10 return π	

Algorithm 3: Local search

  Input: Reduced graph G=(V0,E0), solution π, best solution π⋆, search depth ω	
  Output: Improved solution π	
 1  NoImprove←0      /*Indicate the times of π⋆ being improved*/	
 2 while NoImprove<ω do	
 3    NoImprove←NoImprove+1	
 4    (π,π⋆,NoImprove)←Dynamic_thresholding_search(π,π⋆,NoImprove)    /*Section 3.4.1*/	
 5    (π,π⋆,NoImprove)←Descent_search(π,π⋆,NoImprove)          /*Section 3.4.2*/	
 6 end	
 7 return π	

In this work, three basic move operators (DROP, INSERT and SWAP) are adopted in the thresholding search stage that accepts both equivalent and better solutions. DROP deletes a vertex from the current permutation π and move it to C0; INSERT extends the current permutation π by introducing a new vertex; SWAP deletes a vertex v from the current permutation π and inserts a vertex u into π.

Based on these three move operators, dynamic thresholding search (DTS) adopts both the vertex-based strategy and the prohibition mechanism to balance the exploration and exploitation of the search space. In each search round at this stage, the algorithm first randomly visits all vertices in V0 one by one. For each vertex v considered, the set of candidate move operators is executed depending on whether the vertex is in or out of the permutation π. If the objective value of the obtained solution is not worse than the best solution found ever to a certain threshold, the move operation on the vertex is executed. Otherwise, the operation is rejected. Each time a move is taken, the concerned vertex is marked as tabu and forbidden to be moved again during the next tt iterations ( tt is the tabu tenure). This process continues until all vertices of V0 are traversed.

Algorithm 4: The dynamic thresholding search

  Input: Reduced graph G=(V0,E0), solution π, best solution π⋆, the iterations without improvement NoImprove	
  Output: Solution π, best solution π⋆, the iterations without improvement NoImprove	
 1 Randomly shuffle all vertices in V0	
 2 for each v∈V0 do	
 3   if v∉π then	
 4     if the number of conflicts after inserting v into π is 0 then	
 5           Insert v into π      /* INSERT operator */	
 6      if  f(π)>f(π⋆) then	
 7          π⋆←π, NoImprove←0	
 8       end	
 9     else if v only conflicts with u∈π and has not been involved in any SWAP operation then	
10      Remove u from π and insert v into π     /* SWAP operator */	
11   else	
12     Calculate NM(v) according to Eq. (7)	
13     if NM(v)≠∅ and v has not been involved in any SWAP operation then	
14      Randomly select a vertex u from NM(v)	
15      Remove v from π and insert u into π      /* SWAP operator */	
16     else if f(π)−1>f(π⋆)−δ then	
17      Remove v from π     /* DROP operation */	
18   end	
19 end	
20 return π,π⋆,NoImprove	

(1) If v is outside π (i.e., v∈C0), the candidate move operator set consists of INSERT and SWAP. INSERT is applied first as it improves the solution quality. Then SWAP is applied, which keeps the solution quality unchanged. If neither operator can be applied, DTS just skips v. INSERT can be applied if the number of conflicts of v in π is 0 (i.e., g(π⊕<v,i−(v),i+(v)>)=0). SWAP is applied if v meets two conditions simultaneously: (1) v is not involved in any SWAP operation already taken at the current round; (2) v conflicts with just one vertex u in π (i.e., g(π⊕<v,i−(v),i+(v)>)=1, and can only be swapped with u).

Similar to the INSERT operation in the greedy initialization, we adopt g(π⊕<v,i−(v),i+(v)>) for quick computation during the DTS stage. That is, we only have to update the number of conflicts of the operated vertex v and its neighbor vertices after each move operation, i.e., the updated vertex set is {v}∪{u:(u,v)∈E and u∈C0}. For the INSERT operation, we update the number of conflicts of the vertices neighboring to v and not in π; for the SWAP operation, we update the number of conflicts of v, u, and all vertices neighboring to u and v not in π. Thus, the time complexity of INSERT and SWAP is O(dmax2), where dmax denotes the largest degree of a vertex in the graph.

(2) If v belongs to π, DROP and SWAP are the two candidate operators. SWAP is applied before DROP as SWAP does not degrade the solution quality while DROP does. If neither operation can be applied, the algorithm just skips v. SWAP can be applied only if v satisfies two conditions simultaneously: (1) v was not involved in any SWAP operation already taken at the current round; (2) the set NM(v)⊂C0 of v is non-empty, which is defined as

(7) NM(v)={u:g(π∖{v}⊕<u,i−(u),i+(u)>)=0,c(u,v)=1}.

The vertex u that is to be swapped with v is a random vertex in NM(v).

DROP can be applied if it makes the number of the vertices in π still above the threshold determined by f(π⋆)−δ after the DROP operation, where π⋆ is the best recorded solution and δ (a small positive integer) is a parameter. For the DROP operation, we need to update the number of conflicts of v and all vertices neighboring to v and not in the solution π. The time complexity of DROP is O(dmax2).

Figure 6 shows an example of the dynamic thresholding search stage. To explain this stage, we consider the solution in Fig. 5B as the input solution. For Fig. 5B, C0={a,b,d}, U0=π={e,c} and V0={a,b,c,d,e}. Suppose that the vertices in V0 are randomly shuffled into {a,e,d,b,c}. As shown in Fig. 6A, since the first vertex a is outside π, INSERT and SWAP are the two candidate operators. INSERT is chosen to be applied before SWAP. The INSERT operator cannot be used since the number of conflicts of v is not 0. However SWAP can be applied since c only conflicts with a and a is not in the tabu list. As shown in Fig. 6B, for the second vertex e∈π, SWAP and DROP are the two candidate operators to be considered. SWAP is applied before DROP. SWAP can be applied since in this case NM(e)={b,d} and v is not forbidden by the tabu list. Thus the second vertex e is swapped with a random vertex in NM(e), such as d. After that, the remaining vertices in V0 are evaluated in the same way, while no operators can be applied to them. As a result, the improved solution is π={d,a}.

Algorithm 5: The descent search

  Input: Reduced graph G=(V0,E0), solution π, best solution π⋆, the iterations without improvement NoImprove	
  Output: Solution π, best solution π⋆, the iterations without improvement NoImprove	
 1 while NI(π)≠∅ do	
 2   Randomly select a vertex u from NI(π) and insert it into π	
 3   if f(π)>f(π⋆) then	
 4     π⋆←π, NoImprove←0	
 5   end	
 6 end	
 7 return π,π⋆,NoImprove	

Figure 6 An example of the dynamic thresholding search.

(A) The solution π={e,a} and the set of critical vertices C0={c,b,d} after swapping c with a; (B) the solution π={d,a} and the set of critical vertices C0={c,b,e} after swapping e with d.

The descent search stage

To complement the DTS stage where both equivalent and worse solutions are accepted, the descent search stage is subsequently applied to perform a more intensified examination of candidate solutions. Basically, this stage iteratively selects a conflict-free vertex and inserts it into the solution until such a vertex does not exist anymore.

Figure 7 shows an example of the descent search stage. To explain this stage, we consider the solution in Fig. 6B as the input solution, where C0={c,b,e}, U0=π={d,a} and V0={a,b,c,d,e}. Through computation, NI(π)={e}. As shown in Fig. 7, the vertex e from NI(π) is directly inserted into π and the best solution π⋆ is updated to {d,e,a}.

Figure 7 The local optimal solution π={d,e,a} and the set of critical vertices C0={c,b} of the descent search.

Perturbation procedure

As described in “The Dynamic Thresholding Search Stage”, the threshold search accepts worse solutions that are within a certain quality threshold from the current solution, which relieves the search from the local optimum trap. However, there is a possibility that this strategy may fail. Therefore, we introduce a perturbation strategy that comes into effect when the search falls into a deep stagnation (i.e., the best solution does not change after ω consecutive local search runs). The perturbation strategy incorporates a learning mechanism that gathers move frequencies information from the local search, which is then advantageously used to guide the perturbation.

Algorithm 6 displays the perturbation procedure, which is decomposed into two steps:

Algorithm 6: Learning-based perturbation

  Input: Solution π, the first and the second perturbation strength coefficients β1 and β2	
  Output: Perturbed solution π	
  // Step 1	
 1   L←⌈(β1+β2)×|π|⌉	
 2   A←L vertices in π with the highest move frequencies	
 3   A← sort A in non-increasing order of move frequencies	
 4   NI(π)←∅	
  // Step 2	
 5  for each v∈A do	
 6    Drop the vertex v from π and record its order j in A	
 7    Calculate NI(π) according to Eq. (5)	
 8    if j>⌈β1×(|π|+1)⌉ and NI(π)≠∅ then	
 9     Randomly select a vertex u from NI(π) and insert it into π	
10    end	
11  end	
12 return π	

Step 1: Choose and sort L vertices in π. Choose L vertices in π with the highest move frequencies and sort them in a non-increasing order of the frequencies (lines 1–3, Algorithm 6). The move frequency of each vertex v is the number of times that v has been moved during the local search, which is initially set to 0, and increases by 1 each time v is moved from one set to another.

Step 2: Drop and insert the to-be-perturbed vertices. Each vertex v (v∈A) is dropped, whose order j in A is recorded (line 6, Algorithm 6). If j>⌈β1×(|π|+1)⌉ and NI(π)≠∅, randomly select a vertex u from NI(π) and insert it into π (lines 8–9, Algorithm 6). Recall that NI(π) represents the set of vertices in C0 satisfying the condition g(π⊕⟨v,i−(v)>,i+(v)⟩)=0.

Figure 8 shows an example of the learning-based perturbation applied to a local optimal solution as shown in Fig. 7, where C0={c,b} and U0= π={d,e,a}. Suppose L=2 and the chosen vertices is sorted as A={e,a} according to the move frequencies. The first vertex e is dropped, which leads to an intermediate perturbed solution π={d,a}. Then, the second vertex a is also dropped. Since the order of a is 2, which is more than ⌈β1×(|π|+1)⌉=1 vertex, and NI(π)≠∅, the vertex b is selected randomly from NI(π) and inserted into π, giving the perturbed solution π={b,d}.

Figure 8 An example of the learning-based perturbation.

(A) An intermediate perturbed solution π={d,a} and the set of critical vertices C0={b,c,e}; (B) a perturbed solution π={b,d} and the set of critical vertices C0={a,c,e}.

Recovery procedure

This is a reversed procedure of the reduction procedure. It restores the original graph G=(V,E) from the reduced graph G=(V0,E0) by adding back the removed vertices Ur and the critical vertices Cr. Levy & Low (1988) indicates that the FVS of the original G is C=C0∪Cr. Figure 9 depicts an example that shows how the minimum FVS is determined. In the reduction procedure, Cr={h},Ur={f,g,i,j}. After the search stage, V0={a,b,c,d,e} where C0={b,c},U0={a,d,e}. After the recovery procedure, the FVS of the original G is C=C0∪Cr={b,c,h}.

Figure 9 An example of recovery phase.

(A) Feedback vertex set C0={b,c} for the reduced graph G=(V0,E0); (B) recovered feedback vertex set C={b,c,h}.

Computational complexity and discussion

We consider first the greedy initialization procedure consisting of two stages. The first stage is to initialize the array NI(π), which can be realized in O(|V0|). The complexity of updating NI(π) is O(dmax2). The second stage is to construct the initial solution π, which is bounded by O(|π|×dmax2), and |π| is the size of π. Therefore, the time complexity of the greedy initialization procedure is O(|V0|+|π|×dmax2).

Next, the local search and perturbation procedures in the main loop of IDTS algorithm are considered. In each iteration of the local search, the dynamic threshold search and the descent search stages are performed alternately. The former is realized in O(|π|×dmax+|V0|), and the latter in O(|V0|). Thus, the complexity of the local search procedure is O(K1×(|π|×dmax+|V0|)), where K1 is the number of iterations of the local search. Then, the perturbation procedure can be achieved in O(|π|×(β1+β2×dmax2)), which is much smaller than that of the local search. Therefore, the complexity of one iteration of the main loop of IDTS algorithm is O(K1×(|π|×dmax+|V0|)), and that of SA is O(K2×(dmax2+|V0|)), where K2 is the number of iterations during each temperature period. Therefore, it can be seen that the two complexities are of the same order of magnitude.

Experimental results and comparisons

We test the proposed IDTS algorithm for the FVSP on 71 commonly-used benchmark instances in the literature and 30 large instances generated by this work (“Benchmark Instances”) and compare its results with the state-of-the-art algorithms in “Comparison with State-of-the-Art Results”. In addition to these directed instances, we also present comparative results on directed graphs obtained by a slightly adapted version of the IDTS algorithm (“Comparative Results on Undirected Graphs”). Below, we first present the 101 directed graphs as well as the experiment settings.

Benchmark instances

We use 101 benchmark instances, which are classified into five categories. No optimal solutions are known for the instances of the first to forth categories, while optimal solutions are known for the instances of the fifth category. 1. The first category consists of 40 instances that are randomly generated by Pardalos, Qian & Resende (1998) using the FORTRAN random graph generator mkdigraph.f (http://mauricio.resende.info/data/index.html). The name of these instances is in the form of P|V | − |E|*, where |V|∈{50,100,500,1000} is the number of vertices in the graph, and |E|∈[100,30000] is the number of edges. Given the number of vertices and edges, a graph is built by randomly selecting |E| pairs of vertices as two endpoints of a directed edge. These instances are largely tested in the literature on the FVSP (Galinier, Lemamou & Bouzidi, 2013; Zhou, 2016).

2. The second category is composed of 10 random directed graphs, which are generated in the same way as the first category while the in-degree and out-degree of each vertex are no more than 10. These instances have R|V | − |E|* in their names. The number of vertices |V | is in the interval [100, 3,000], and the number of edges |E| in [500, 15,000].

3. The third category contains 10 artificially generated scale-free instances. These instances are generated by this work through the “powerlaw_cluster_graph” function of the “NetworkX” package, which is based on the algorithm proposed by Holme & Kim (2002). These instances are named as S|V | − |E|*, where |V | is in the interval [500, 3,000] and |E| is in [4,900, 29,900].

4. The fourth category is composed of 10 real-world instances from the Stanford large network dataset collection (http://snap.stanford.edu/data/). Nine of these instances are snapshots of the Gnutella peer-to-peer file sharing network. The remaining instance is a temporal network representing Wikipedia users editing each other’s Talk page. The number of vertices |V | is in the interval [6,301, 1,140,149], and the number of edges |E| in [20,777, 7,833,140].

5. The fifth category is composed of the 31 classical (easy) ISCAS89 benchmark instances which are from digital sequential circuits (Brglez, Bryan & Kozminski, 1989). These instances have s* in their names, where the number of vertices is in the range of [3, 1,728], and the number of edges in the range of [4, 32,774]. These instances, whose optima are known, are largely tested in the literature on the FVSP (Levy & Low, 1988; Lin & Jou, 1999; Orenstein, Kohavi & Pomeranz, 1995).

Experiment settings

The IDTS algorithm is programmed in C++ and compiled by GNU g++ 4.1.2 with the -O3 flag. Experiments are carried out on a computer with an Intel(R) Core(TM)2 Duo CPU T7700 2.4 GHz processor with 2 GB RAM running Ubuntu CentOS Linux release 7.9.2009 (Core).

Parameters

The IDTS algorithm requires five parameters: the maximum non-improving iteration depth ω of local search, the tabu tenure tt, the first perturbation strength coefficient β1, the second perturbation strength coefficient β2 and the thresholding coefficient δ. To tune these parameters, the “IRACE” package (López-Ibáñez et al., 2016) was adopted to automatically recognize a group of appropriate values for eight representative instances (with 50–30,000 vertices), and its budget was set to 200 runs under a cutoff time described in “Stopping Conditions”. Table 1 presents both considered values and final tuned values of these parameters.

Table 1 Settings of important parameters.

Parameters	Section	Description	Considered value	Final value	
ω	3.4	Search depth of the local search	{10, 20, 30, 40, 50, 60, 70}	20	
tt	3.4.1	Tabu tenure	{1, 2, 3, 4, 5, 6}	1	
δ	3.4.1	Thresholding coefficient ( 50≤|V|≤100)	{1, 2, 3, 4, 5, 6}	1	
		Thresholding coefficient ( 500≤|V|≤ 1,000)	{1, 2, 3, 4, 5, 6}	4	
		Thresholding coefficient (1,000 <|V|≤ 3,000)	{5, 10, 15, 20, 25, 30}	10	
		Thresholding coefficient (3,000 <|V|)	{5, 10, 15, 20, 25, 30}	20	
β1	3.5	The first perturbation strength coefficient	{0.02, 0.04, 0.06, 0.08, 0.1, 0.2}	0.04	
β2	3.5	The second perturbation strength coefficient	{0.1, 0.2, 0.3, 0.4, 0.5, 0.6}	0.3	

These parameter values can be considered to form the default setting of the IDTS algorithm and were consistently used for our experiments to ensure a meaningful comparative study. By fine-tuning some parameters on an instance-by-instance basis, it would be possible to obtain better results.

Reference algorithms

Three state-of-the-art FVSP algorithms are adopted as reference methods to evaluate the IDTS algorithm for directed graphs. (1) Simulated annealing algorithm (SA) for the first category (Galinier, Lemamou & Bouzidi, 2013);

(2) Algorithm combining the reduction procedure and the simulated annealing algorithm (Red+SA) for the first category (Galinier, Lemamou & Bouzidi, 2013), the re-implemented Red+SA (Re-Red+SA) for the categories two to five;

(3) Belief propagation-guided decimation algorithm (BPD) (Zhou, 2016).

Among them, the codes of BPD were kindly provided by its author, and were run by us under the same experimental conditions as for the IDTS algorithm for a fair comparison. We also carefully re-implemented the Red+SA algorithm (Galinier, Lemamou & Bouzidi, 2013), since its codes are unavailable. We used the re-implemented Red+SA algorithm (Re-Red+SA) to solve the instances of categories two to fifth and cited the results in Galinier, Lemamou & Bouzidi (2013) for the first category. Galinier, Lemamou & Bouzidi (2013) used a computer (Intel(R) Core(TM)) 2 CPU T8300 2.4 GHz with 2 GB of RAM, which is comparable to our Intel computer running at 2.40 GHz.

Stopping conditions

Cutoff time of each run. Reference algorithms BPD (Zhou, 2016) and SA (Galinier, Lemamou & Bouzidi, 2013) have different stopping conditions. Thus, we adopted these average computation times as the cutoff times for our IDTS algorithm for fairness. Following Galinier, Lemamou & Bouzidi (2013), for the instances of the first category, the cutoff time is set to 0.03 to 0.07 s for n=50, 0.06 to 0.34 s for n=100, 1.8 to 5.2 s for n=500, 11 to 25.5 s for n = 1,000. For the second and third categories, the cutoff time is set to 1,200 s. For the fourth category, the cutoff time is set to 6,000 s for all compared algorithms. For the easy fifth category, the cutoff time is set to 15 s.

Normal test. Following Galinier, Lemamou & Bouzidi (2013), we firstly ran our IDTS algorithm 30 times per instance with the above cutoff time.

Relaxed test. The SA algorithm (the Red+SA algorithm without the reduction procedure) (Galinier, Lemamou & Bouzidi, 2013), was run 1,000 times on each instance. Under this condition, it reported the currently best objective values for the benchmark instances of the first category. Like Galinier, Lemamou & Bouzidi (2013), we also ran IDTS 1,000 times on each instance of the first category under the same stopping conditions.

Comparison with state-of-the-art results

Comparison of the results on the first-category instances

Table 2 displays the results of the Red+SA, BPD, and IDTS algorithms on the commonly-used 40 instances of the first category in the literature. The first three columns reveal the name, the number of vertices and the number of edges of each instance. Columns 4–7 provide the results of the Red+SA on each instance: the best objective value (Best) over 30 independent runs, the worst result (Worst), the average result (Avg), and the cutoff time (in seconds). Columns 8–15 report the results of the the BPD and IDTS algorithm: the best, worst, average objective values and the average computation time (in seconds) to obtain the best result ( t(s)). The last two columns ( Δ1 and Δ2) indicate the difference between our best results (Best) and those of Red+SA and BPD (a negative value indicates an improved result). The row “p-value” is given to verify the statistical significance of the comparison between IDTS and the reference algorithms, which came from the non-parametric Friedman test applied to the best, worst and average values of IDTS and reference algorithms. A p-value less than 0.05 indicates a statistically significant difference.

Table 2 Comparative results of IDTS with state-of-the-art algorithms on the 40 benchmark instances of the first category in normal test (30 independent runs).

Instance	|V |	|E|	Red+SA (Galinier, Lemamou & Bouzidi, 2013)	BPD (Zhou, 2016)	IDTS	Δ1	Δ2	
			Best	Worst	Avg	cutoff(s)	Best	Worst	Avg	t(s)	Best	Worst	Avg	t(s)			
P50-100	50	100	3	3	3	0.03	3	4	3.2	0.01	3	3	3	0.01	0	0	
P50-150	50	150	9	9	9	0.03	9	11	9.2	0.02	9	9	9	0.01	0	0	
P50-200	50	200	13	13	13	0.03	13	14	13.1	0.04	13	13	13	0.01	0	0	
P50-250	50	250	17	17	17	0.04	17	19	17.5	0.06	17	17	17	0.01	0	0	
P50-300	50	300	19	19	19	0.04	19	22	19.5	0.08	19	19	19	0.01	0	0	
P50-500	50	500	28	28	28	0.05	28	31	29.3	0.16	28	28	28	0.01	0	0	
P50-600	50	600	31	32	31.4	0.07	31	34	32.6	0.20	31	31	31	0.01	0	0	
P50-700	50	700	33	33	33	0.05	33	34	33.3	0.24	33	33	33	0.01	0	0	
P50-800	50	800	34	35	34.1	0.07	34	38	35.4	0.29	34	34	34	0.01	0	0	
P50-900	50	900	36	36	36	0.04	36	38	36.4	0.32	36	36	36	0.01	0	0	
P100-200	100	200	9	9	9	0.06	9	11	10.0	0.03	9	9	9	0.01	0	0	
P100-300	100	300	17	17	17	0.08	17	18	17.3	0.11	17	17	17	0.01	0	0	
P100-400	100	400	23	23	23	0.1	23	25	23.5	0.18	23	23	23	0.01	0	0	
P100-500	100	500	32	33	32.3	0.14	33	37	34.4	0.33	32	32	32	0.03	0	−1	
P100-600	100	600	37	37	37	0.15	37	41	38.9	0.45	37	37	37	0.56	0	0	
P100-1000	100	1,000	53	54	53.2	0.27	54	57	55.5	1.07	53	53	53	0.15	0	−1	
P100-1100	100	1,100	54	55	54.8	0.23	55	59	55.8	1.26	54	55	54.7	0.11	0	−1	
P100-1200	100	1,200	57	57	57	0.29	58	62	59.4	1.40	57	57	57	0.23	0	−1	
P100-1300	100	1,300	60	60	60	0.31	61	66	62.7	1.55	60	60	60	0.02	0	−1	
P100-1400	100	1,400	61	61	61	0.34	62	65	63.0	1.80	61	61	61	0.09	0	−1	
P500-1000	500	1,000	31	33	32.1	1.78	31	35	32.3	0.93	31	31	31	0.05	0	0	
P500-1500	500	1,500	63	66	65.1	2.35	65	69	66.9	3.44	63	64	63.8	1.36	0	−2	
P500-2000	500	2,000	102	106	104	2.53	104	108	105.5	7.27	101	105	102.8	0.39	−1	−3	
P500-2500	500	2,500	133	138	135.5	2.62	137	142	140.0	12.88	132	137	135.1	1.62	−1	−5	
P500-3000	500	3,000	163	168	165.4	2.94	165	172	168.2	19.04	163	169	164.9	1.91	0	−2	
P500-5000	500	5,000	237	241	239.2	3.95	240	247	243.8	54.91	237	242	240.1	0.87	0	−3	
P500-5500	500	5,500	252	256	253.8	4.04	254	260	256.8	68.20	252	257	254.7	2.68	0	−2	
P500-6000	500	6,000	265	270	267.6	4.64	268	273	270.2	81.39	264	270	267.6	2.62	−1	−4	
P500-6500	500	6,500	277	283	278.9	4.79	279	287	283.0	93.33	276	282	278.5	2.91	−1	−3	
P500-7000	500	7,000	287	292	288.9	5.2	288	297	292.5	109.96	287	292	288.7	5.15	0	−1	
P1000-3000	1,000	3,000	128	135	131.2	11.53	130	136	133.4	12.84	128	132	129.9	10.98	0	−2	
P1000-3500	1,000	3,500	163	169	166.5	12.34	163	172	167.4	20.10	162	167	164.3	5.44	−1	−1	
P1000-4000	1,000	4,000	194	201	197.3	12.93	195	203	199.3	29.03	193	198	195.5	9.27	−1	−2	
P1000-4500	1,000	4,500	230	237	233.5	12.21	230	238	233.2	40.14	229	237	231.5	8.26	−1	−1	
P1000-5000	1,000	5,000	263	269	265.7	11.7	261	268	263.8	52.53	261	267	263.2	6.79	−2	0	
P1000-10000	1,000	10,000	472	479	475.4	13.45	474	483	478.0	243.24	472	479	475.1	11.41	0	−2	
P1000-15000	1,000	15,000	582	588	584.9	16.73	584	597	589.4	508.33	580	589	585.6	15.31	−2	−4	
P1000-20000	1,000	20,000	652	660	656.1	20.29	654	665	660.0	840.20	652	660	657.3	13.06	0	−2	
P1000-25000	1,000	25,000	701	707	704.5	24.76	704	716	710.0	1,224.58	700	708	704.4	18.73	−1	−4	
P1000-30000	1,000	30,000	741	747	744	25.47	745	754	749.9	1,698.77	741	747	744.1	19.82	0	−4	
#Better			0	5	2		0	0	0								
#Equal			30	20	14		16	0	0								
#Worse			10	15	24		24	40	40								
p-value			1.73E−03	1.63E−02	1.72E−02		9.63E−07	2.54E−10	2.54E−10								
Note: The bold numbers in the table highlight the dominating results between the compared algorithms in terms of Best, Worst and Avg values.

Moreover, the rows #Better, #Equal, and #Worse indicate the number of instances for which Red+SA and BPD obtained a better, equal, and worse result compared to the IDTS algorithm for each performance indicator. The bold entries highlight the dominating results between the compared algorithms in terms of Best, Worst and Avg values.

We notice from Table 2 that IDTS performs satisfactorily and dominates the Red+SA algorithm by obtaining better results (Best) for 10 instances (see negative entries in column Δ1) and equally-good results for the rest 30 instances. IDTS also gets better results in terms of the worst and average results. As for BPD, in terms of the best results, IDTS obtains 16 better (see negative entries in column Δ2) and 24 equal values; in terms of the worst and average results, IDTS obtains better values for all instances. The small p-values (<0.05) confirm the statistical significance of the reported differences between IDTS and the reference algorithms.

In Galinier, Lemamou & Bouzidi (2013), SA (i.e., the Red+SA algorithm without the reduction procedure) reported several improved results over 1,000 runs compared to the results of Red+SA in Table 2. Similarly, the IDTS algorithm was run 1,000 times, and the comparative results of SA and IDTS are shown in Table 3, where the last column ( Δ) shows the difference between the best results of IDTS (Best) and those of SA (a negative value indicates a better result). It reveals that IDTS further improves the results of SA and discovers six record-breaking results (indicated in bold) for the instances P500-2000, P500-2500, P1000-3000, P1000-3500, P1000-4000 and P1000-5000.

Table 3 Comparative results of IDTS with state-of-the-art algorithm on the 40 benchmark instances of the first category in relaxed test (1,000 independent runs).

Instance	|V |	|E|	SA	IDTS	Δ	
			Best	t(s)	Best	t(s)		
P500-2000	500	2,000	102	–	100	1.07	−2	
P500-2500	500	2,500	133	–	131	1.74	−2	
P1000-3000	1,000	3,000	128	–	127	8.39	−1	
P1000-3500	1,000	3,500	163	–	161	7.72	−2	
P1000-4000	1,000	4,000	194	–	191	7.61	−3	
P1000-5000	1,000	5,000	259	–	258	9.81	−1	
Note: The bold numbers in the table highlight the dominating results between the compared algorithms in terms of Best, Worst and Avg values.

Comparison of the results on the second-category instances

Table 4 shows the comparative results between IDTS and the reference algorithms on the 10 instances of the second-category. In terms of the best results, IDTS dominates Re-Red+SA by obtaining better values for all instances, and BPD by obtaining 7 better, and 3 equal results. On the other hand, IDTS significantly outperforms Re-Red+SA and BPD in terms of the worst and average results by obtaining better or equal results for all instances (except for R3000-15000). The small p-values (<0.05) indicate that there are significant differences between our best results and those of the two reference algorithms Re-Red+SA (p-value = 1.60E−3) and BPD (p-value = 8.20E−03).

Table 4 Comparative results of IDTS with state-of-the-art algorithms on the 10 benchmark instances of the second category in normal test (30 independent runs).

Instance	|V |	|E|	Re-Red+SA	BPD (Zhou, 2016)	IDTS	Δ1	Δ2	
			Best	Worst	Avg	t(s)	Best	Worst	Avg	t(s)	Best	Worst	Avg	t(s)			
R100-500	100	500	34	41	36.6	0.00	30	33	31.0	0.32	30	30	30.0	0.03	−4	0	
R200-1000	200	1,000	66	74	70.3	0.01	58	62	60.2	2.04	57	57	57.0	0.37	−9	−1	
R500-2500	500	2,500	143	167	156.8	3.09	139	144	141.1	13.86	136	138	136.7	15.67	−7	−3	
R800-4000	800	4,000	264	287	274.8	0.47	220	227	224.1	35.78	216	218	217.1	37.9	−48	−4	
R1000-5000	1,000	5,000	302	332	319.0	11.64	262	276	267.6	59.05	262	267	264.7	30.14	−40	0	
R1250-7500	1,250	7,500	474	518	500.5	40.74	406	411	409.1	139.70	403	411	406.6	98.62	−71	−3	
R1500-9000	1,500	9,000	578	625	599.6	68.17	480	490	484.3	200.43	477	486	481.7	169.66	−101	−3	
R1750-10500	1,750	10,500	658	727	694.9	131.02	561	570	566.0	270.63	561	569	564.2	287.72	−97	0	
R2000-12000	2,000	12,000	726	808	763.6	163.09	640	650	644.5	351.42	638	648	643.5	340.99	−88	−2	
R3000-15000	3,000	15,000	835	874	850.9	508.32	786	801	791.5	326.78	785	801	792.7	567.09	−50	−1	
#Better			0	0	0		0	0	1								
#Equal			0	0	0		3	2	0								
#Worse			10	10	10		7	8	9								
p-value			1.60E−03	1.60E−03	1.60E−03		8.20E−03	4.70E−03	1.14E−02								
Note: The bold numbers in the table highlight the dominating results between the compared algorithms in terms of Best, Worst and Avg values.

Furthermore, Fig. 10 summarizes the performance of the IDTS algorithm with that of the Re-Red+SA and BPD algorithms on these instances. Figure 10A presents the relationship between the number of vertices and the best FVS size (the best objective value over 30 runs). Figure 10B shows the relationship between the number of vertices and the average computation time. One observes that the FVS size increases linearly while the average computation time increases exponentially with the increase of the number of vertices.

Figure 10 Comparisons of IDTS (in red) with BPD (in black) and SA (in blue).

(A) The best objective value; (B) the average computation time.

Comparison of the results on the third-category instances

Table 5 presents the comparative results of IDTS with the reference algorithms Re-Red+SA and BPD for the instances of the third category. As shown in Table 5, IDTS outperforms Re-Red+SA by obtaining better results for all instances in terms of the best, worst and average results. Compared with BPD, IDTS obtains seven better, two equal, and one worse values in terms of the best results; seven better, one equal, and two worse values in terms of the worst results; six better, one equal, and three worse values in terms of the average results. Finally, the p-values smaller than 0.05 indicate IDTS significantly dominates each reference algorithm in terms of the best results.

Table 5 Comparative results of IDTS with state-of-the-art algorithms on the 10 benchmark instances of the third category in normal test (30 independent runs).

Instance	|V |	|E|	Re-Red+SA	BPD (Zhou, 2016)	IDTS	Δ1	Δ2	
			Best	Worst	Avg	t(s)	Best	Worst	Avg	t(s)	Best	Worst	Avg	t(s)			
S500-4900	500	4,900	177	184	179.6	27.11	175	182	178.7	40.02	172	175	173.5	45.26	−5	−3	
S1000-9900	1,000	9,900	339	349	343.6	30.08	328	335	331.0	174.32	325	329	329.5	93.55	−14	−3	
S1250-12400	1,250	12,400	405	412	408.0	115.44	397	405	400.2	273.29	394	399	396.5	233.10	−11	−3	
S1500-14900	1,500	14,900	487	497	492.9	92.03	479	489	484.8	381.63	476	485	480.9	374.39	−11	−3	
S1750-17400	1,750	17,400	567	582	573.6	184.90	556	563	559.7	542.47	553	561	556.0	537.69	−14	−3	
S2000-19900	2,000	19,900	658	674	664.7	261.05	630	639	633.4	698.89	627	640	635.3	627.08	−31	−3	
S2250-22400	2,250	22,400	723	734	730.2	321.75	703	713	707.3	634.98	700	709	704.2	546.53	−23	−3	
S2500-24900	2,500	24,900	810	825	817.0	374.91	783	792	788.1	691.97	783	792	788.1	673.39	−27	0	
S2750-27400	2,750	27,400	887	906	896.6	535.72	855	863	857.9	804.53	855	862	859.3	700.99	−32	0	
S3000-29900	3,000	29,900	981	1,004	991.5	650.09	934	945	940.8	895.73	938	950	948.0	1,111.52	−43	4	
#Better			0	0	0		1	2	3								
#Equal			0	0	0		2	1	1								
#Worse			10	10	10		7	7	6								
p-value			1.60E−03	1.60E−03	1.60E−03		3.39E−02	9.56E−02	3.17E−01								
Note: The bold numbers in the table highlight the dominating results between the compared algorithms in terms of Best, Worst and Avg values.

Comparison of the results on the fourth-category instances

The comparative results of IDTS and the reference algorithms Re-Red+SA and BPD on the fourth category are summarized in Table 6. It can be seen that IDTS outperforms the reference algorithms for the instances of the fourth-category. Compared with Re-Red+SA, IDTS obtains nine better and one equal results in terms of the best results, and better worst and average values for all instances. Compared with BPD, IDTS obtains five better, four equal, and one worse values in terms of the best results; four better, one equal and five worse values in terms of the worst and average results. The p-value of 2.70E−03 between IDTS and Re-Red+SA in terms of the best results indicate that there are significant differences between their results.

Table 6 Comparative results of IDTS with state-of-the-art algorithms on the 10 benchmark instances of the forth category in normal test (30 independent runs).

Instance	|V |	|E|	Re-Red+SA	BPD (Zhou, 2016)	IDTS	Δ1	Δ2	
			Best	Worst	Avg	t(s)	Best	Worst	Avg	t(s)	Best	Worst	Avg	t(s)			
p2p-Gnutella04	10,879	39,994	573	583	577.1	2,078.25	557	569	563.4	283.23	557	571	565.3	858.71	−16	0	
p2p-Gnutella05	8,846	31,839	367	380	371.8	1,077.66	367	374	370.6	212.56	365	370	368.0	215.36	−2	−2	
p2p-Gnutella06	8,717	31,525	406	416	410.9	1,080.73	401	409	404.6	229.35	400	405	402.9	235.74	−6	−1	
p2p-Gnutella08	6,301	20,777	193	201	197.9	357.42	196	202	198.8	216.42	193	198	196.4	223.64	0	−3	
p2p-Gnutella09	8,114	26,013	260	268	265.4	611.79	261	266	262.9	184.28	259	263	261.0	195.57	−1	−2	
p2p-Gnutella24	26,518	65,369	797	811	803.1	5,646.59	784	797	790.2	304.86	783	799	793.1	811.81	−14	−1	
p2p-Gnutella25	22,687	54,705	621	638	626.1	4,676.26	607	621	614.6	234.32	607	623	617.0	723.31	−14	0	
p2p-Gnutella30	36,682	88,328	950	970	961.4	5,830.20	910	926	916.7	374.05	913	931	924.7	1,538.83	−37	3	
p2p-Gnutella31	62,586	147,892	1,416	1,443	1,429.2	5,739.32	1,247	1,264	1,254.9	522.33	1,247	1,280	1,262.6	3,160.59	−169	0	
wiki-talk-temporal	1,140,149	7,833,140	179,486	179,503	179,494.5	3.25	179,472	179,472	179,472.0	5,138.57	179,472	179,472	179,472.0	3.02	−14	0	
#Better			0	0	0		1	5	5								
#Equal			1	0	0		4	1	1								
#Worse			9	10	10		5	4	4								
p-value			2.70E−03	1.60E−03	1.60E−03		1.03E−01	7.39E−01	7.39E−01								
Note: The bold numbers in the table highlight the dominating results between the compared algorithms in terms of Best, Worst and Avg values.

Results on the ISCAS89 benchmark instances

Table 7 shows the results of IDTS on the classical ISCAS89 benchmark instances. The instances with known optimal values were solved exactly by the branch and bound algorithm (H8WR) combined with eight reduction operations (Lin & Jou, 1999) and indicated by asterisks (*). It can be observed that IDTS can easily reach the optimal solutions for these instances, while Re-Red+SA and BPD miss the two optimal solutions indicated in boldface.

Table 7 Results of IDTS on the ISCAS89 benchmark instances.

Instance	|V |	|E|	Optimal value	Re-Red+SA	BPD (Zhou, 2016)	IDTS	
				Best	t(s)	Best	t(s)	Best	t(s)	
s1196*	18	20	0	0	0.01	0	0.01	0	0.01	
s1238*	18	20	0	0	0.01	0	0.01	0	0.01	
s13207*	669	3,406	59	59	0.01	59	10.17	59	0.4	
s1423*	74	1,694	21	21	0.01	21	2.02	21	0.02	
s1488*	6	30	5	5	0.01	5	0.01	5	0.01	
s1494*	6	30	5	5	0.01	5	0.01	5	0.01	
s15850*	597	14,925	88	88	0.09	88	284.25	88	0.11	
s208*	8	28	0	0	0.01	0	0.01	0	0.01	
s27*	3	4	1	1	0.01	1	0.01	1	0.01	
s298*	14	56	1	1	0.01	1	0.01	1	0.01	
s344*	15	74	5	5	0.01	5	0.01	5	0.01	
s349*	15	74	5	5	0.01	5	0.01	5	0.01	
s35932*	1,728	4,475	306	306	0.01	306	37.94	306	6.41	
s382*	21	131	9	9	0.01	9	0.01	9	0.01	
s38417*	1,636	32,774	374	375	1.49	380	1,384.78	374	5.24	
s38584*	1,452	16,880	292	294	6.02	293	569.37	292	14.9	
s386*	6	30	5	5	0.01	5	0.01	5	0.01	
s400*	21	131	9	9	0.01	9	0.01	9	0.01	
s420*	16	72	0	0	0.01	0	0.01	0	0.01	
s444*	21	131	9	9	0.01	9	0.01	9	0.01	
s510*	6	30	5	5	0.01	5	0.01	5	0.01	
s526*	21	123	3	3	0.01	3	0.01	3	0.01	
s526n*	21	123	3	3	0.01	3	0.01	3	0.01	
s5378*	179	1,200	30	30	0.01	30	0.93	30	0.01	
s641*	19	100	7	7	0.01	7	0.01	7	0.01	
s713*	19	100	7	7	0.01	7	0.01	7	0.01	
s820*	5	20	4	4	0.01	4	0.01	4	0.01	
s832*	5	20	4	4	0.01	4	0.01	4	0.01	
s838*	32	160	0	0	0.01	0	0.01	0	0.01	
s9234*	228	2,680	53	53	0.01	53	3.77	53	0.02	
s953*	29	150	5	5	0.01	5	0.01	5	0.01	
Note: The bold numbers in the table highlight the dominating results between the compared algorithms in terms of Best, Worst and Avg values; the instances with known optimal values are indicated by asterisks (*).

Comparative results on undirected graphs

To make our IDTS algorithm applicable to undirected graphs, we modified the neighborhood condition that the number of conflicts equals 0 (as described in “Preliminaries”) to the constraint that for any vertex v∈π, there is at most one neighbor vertex u∈π of v in front of v.

For our comparative study, we carefully re-implemented the SALS algorithm (Qin & Zhou, 2014) as its codes are unavailable. We regenerated 20 instances of the same characteristics using the generation method of Qin & Zhou (2014). These instances have ER* or RR* in their names, where the number of vertices is 100,000, and the number of edges is in the range of [100,000, 1,000,000]. The cutoff time is set to 6,000 s and both algorithms were run 30 times per instance.

Table 8 displays the results of the SALS and IDTS algorithms on the 20 regenerated instances. Columns 1–3 show the name, the number of vertices and the number of edges of each instance. Columns 4–11 respectively provide the results of the SALS and the IDTS on each instance: the best objective value (Best) over 30 independent runs, the worst result (Worst), the average result (Avg), and the average computation time (in seconds) to obtain the best result ( t(s)). The last column ( Δ) indicates the differences between our best results (Best) and those of SALS (a negative value indicates an improved result). The row “p-value” is given to verify the statistical significance of the comparison between IDTS and the reference algorithm, which came from the non-parametric Friedman test applied to the best, worst and average values of the two compared algorithms.

Table 8 Comparative results of IDTS with state-of-the-art algorithm SALS (Qin & Zhou, 2014) on the 20 undirected instances in normal test (30 independent runs).

Instance	|V |	|E|	SALS	IDTS	Δ	
			Best	Worst	Avg	t(s)	Best	Worst	Avg	t(s)		
ER1	100,000	100,000	12,105	12,178	12,149.8	3,473.67	7,534	7,658	7,593.0	5,041.77	−4,571	
ER2	100,000	200,000	43,580	43,756	43,682.1	3,037.78	25,185	25,633	25,374.3	5,456.73	−18,395	
ER3	100,000	300,000	59,755	59,886	59,821.1	3,035.21	38,075	38,325	38,176.3	6,000.83	−21,680	
ER4	100,000	400,000	68,915	69,066	68,985.3	2,957.06	47,447	47,640	47,537.8	5,589.16	−21,468	
ER5	100,000	500,000	74,774	74,873	74,828.0	3,132.84	54,050	54,403	54,221.0	5,501.83	−20,724	
ER6	100,000	600,000	78,799	78,883	78,854.9	3,512.93	58,929	59,389	59,231.0	5,433.78	−19,870	
ER7	100,000	700,000	81,719	81,809	81,770.0	3,535.40	63,269	63,352	63,323.7	6,000.12	−18,450	
ER8	100,000	800,000	83,910	84,034	83,994.2	2,593.17	66,304	66,912	66,502.3	5,652.81	−17,606	
ER9	100,000	900,000	85,646	85,765	85,723.5	3,352.80	69,031	69,562	69,208.5	5,760.04	−16,615	
ER10	100,000	1,000,000	87,082	87,153	87,123.8	3,564.80	71,311	71,707	71,503.5	5,816.93	−15,771	
RR1	100,000	100,000	4	4	4.0	610.92	4	4	4.0	1,343.69	0	
RR2	100,000	200,000	49,228	49,331	49,294.4	2,874.34	35,081	35,350	35,214.8	5,439.20	−14,147	
RR3	100,000	300,000	64,327	64,389	64,357.5	3,167.82	47,042	47,578	47,264.5	5,461.19	−17,285	
RR4	100,000	400,000	72,185	72,306	72,255.7	2,607.66	54,771	55,606	55,041.5	5,578.65	−17,414	
RR5	100,000	500,000	77,136	77,250	77,221.6	3,085.60	60,253	60,776	60,489.8	5,533.04	−16,883	
RR6	100,000	600,000	80,583	80,691	80,653.8	3,591.49	64,484	64,971	64,695.0	5,660.65	−16,099	
RR7	100,000	700,000	83,152	83,208	83,180.6	3,382.05	67,609	67,634	67,624.7	5,905.53	−15,543	
RR8	100,000	800,000	85,065	85,134	85,108.8	3,347.80	70,292	70,394	70,342.3	5,564.86	−14,773	
RR9	100,000	900,000	86,608	86,665	86,635.4	3,355.52	72,243	72,938	72,610.5	5,616.01	−14,365	
RR10	100,000	1,000,000	87,829	87,907	87,881.2	3,527.91	74,250	74,828	74,580.0	5,553.27	−13,579	
#Better			0	0	0							
#Equal			1	1	1							
#Worse			19	19	19							
p-value			1.31E−05	1.31E−05	1.31E−05							
Note: The bold numbers in the table highlight the dominating results between the compared algorithms in terms of Best, Worst and Avg values.

Moreover, the rows #Better, #Equal, and #Worse indicate the number of instances for which SALS obtained a better, equal, and worse result compared with the IDTS algorithm for each performance indicator. The bold entries highlight the dominating results between the compared algorithms in terms of the Best, Worst and Avg values.

The results indicate that our algorithm dominates the SALS algorithm (Qin & Zhou, 2014) by obtaining 19 better and one equal value in terms of the best, worst and average results. The small p-values (< 0.05) indicate that there are significant differences between our results and those of the reference algorithm SALS. This experiment demonstrates that the proposed algorithm is not only competitive for directed graphs, but performs very well for the undirected case of the problem as well.

Analysis

This section conducts extra tests to analyze the advantages of two important components of the proposed IDTS algorithm: the thresholding coefficient and the perturbation strategy.

Effects of the thresholding coefficient

IDTS adopts the thresholding strategy illustrated in “The Dynamic Thresholding Search Stage” to search both equivalent and better solutions. The oscillation between equivalent and better zones follows the increasing/decreasing of the thresholding coefficient δ (≥1) with an adjustment value. Thus, we analyze the effects of the thresholding coefficient by testing five candidate adjustment values: 1, 2, 3, 4, 5 for the instances with |V|≤ 1,000 and 5, 10, 15, 20, 25 for the instances with |V|> 1,000 (the higher the value, the larger the oscillation between equivalent and better solutions).

Figure 11 shows the Box and whisker plots of the results on eight representative instances with different number of vertices. Where the X-axis refers to the tested adjustment values and the Y-axis stands for the best objective values obtained. As a complement, we also calculate the p-values for each tested instance. Results are from 20 independent runs of each instance with a cutoff time as described in “Stopping Conditions” per run. We observe that the adjustment values affect the performance of IDTS algorithm greatly for most instances except two instances (P500-7000 and p2p-Gnutella25). Moreover, then IDTS algorithm with the adjustment value 1 performs the best on instances with a number of vertices ( 50≤|V|≤100), with the adjustment value 4 on instances with a number of vertices ( 500≤|V|≤1,000 1,000), with the adjustment value 10 on instances with a number of vertices ( 1,000<|V|≤3,000 3,000), and with the adjustment value 20 on instances with a number of vertices ( |V|>3,000 3,000). Finally, it is noted that the results of this experiment are consistent with the intuitive understanding that the higher the adjustment value, the more frequent the oscillation of the search between current configuration and new configuration. That is, large instances require large adjustment values to explore more new areas, while small instances require small adjustment values to fully explore each search area.

Figure 11 Effects of the increase/decrease value of the thresholding coefficient.

(A) P50-600 (p-value = 1.68E−02); (B) P100-1100 (p-value = 9.37E−05); (C) P500-7000 (p-value = 2.71E−01); (D) P1000-30000 (p-value = 4.35E−05); (E) R1250-7500 (p-value = 4.53E−03); (F) S1250-12400 (p-value = 4.05E−04); (G) p2p-Gnutella24 (p-value = 1.37E−02); (H) p2p-Gnutella25 (p-value = 9.11E−01).

Effects of the perturbation operation

To evaluate the perturbation strategy of the proposed algorithm, we create two algorithmic variants (IDTS1 and IDTS2) where the perturbation strategy visits only feasible solutions. For IDTS, the perturbation first drops β1×|π| vertices with the highest move frequency, and then applies both DROP and INSERT moves to the next β2×|π| most frequently displaced vertices. For IDTS1, the perturbation strategy is disabled (i.e., by removing the line 7 in Algorithm 1). For IDTS2, the perturbation strategy only adopts the DROP move (by disabling lines 8–10 in Algorithm 6). A total of 20 relatively difficult instances are selected as per the results provided in Tables 2 and 3, that is, their best results could not be achieved by all algorithms. We ran IDTS, IDTS1 and IDTS2 10 times to solve each selected instance under the same stopping conditions as before.

Table 9 displays the experimental results. The rows #Better, #Equal, and #Worse show the number of instances for which IDTS1 and IDTS2 achieved a better, equal, or worse result than the IDTS algorithm for each performance indicator.

Table 9 Evaluation of the perturbation strategy.

Instance	IDTS	IDTS1	IDTS2	Δ1	Δ2	
	Best	Worst	Avg	t(s)	Best	Worst	Avg	t(s)	Best	Worst	Avg	t(s)			
P100-1100	54	55	54.7	0.13	54	55	54.8	0.06	54	55	54.8	0.04	0	0	
P500-1500	63	64	63.8	0.79	63	64	63.9	0.56	63	64	63.9	0.67	0	0	
P500-2000	102	104	102.7	1.36	102	104	102.9	1.67	102	104	102.6	1.22	0	0	
P500-2500	132	136	134.7	1.37	133	136	134.8	1.23	134	136	135.4	1.51	−1	−2	
P500-3000	163	167	164.7	1.87	163	166	164.8	1.84	163	166	164.9	1.21	0	0	
P500-5000	237	242	240.4	2.52	239	242	240.1	2.11	237	241	239.1	2.68	−2	0	
P500-5500	252	256	253.8	2.46	253	258	255.5	2.85	252	256	253.8	2.67	−1	0	
P500-6000	264	270	267.7	2.64	266	270	267.7	2.79	265	269	267.0	3.22	−2	−1	
P500-6500	278	281	279.6	1.46	278	281	279.3	2.86	278	280	278.6	3.03	0	0	
P500-7000	287	291	289.4	2.72	287	292	289.8	4.64	288	291	289.6	2.63	0	−1	
P1000-3000	128	131	129.3	3.79	128	131	129.5	3.13	128	131	130.5	6.41	0	0	
P1000-3500	162	164	163.3	9.80	163	166	164.3	9.51	163	167	165.6	6.33	−1	−1	
P1000-4000	194	197	194.7	7.48	194	196	195.3	3.96	195	197	196.4	9.34	0	−1	
P1000-4500	228	234	231.2	5.57	228	236	232.1	10.89	228	234	231.4	7.62	0	0	
P1000-5000	263	265	264.0	9.93	263	266	263.9	11.19	264	267	265.1	7.72	0	−1	
P1000-10000	473	479	475.8	12.72	474	483	477.7	10.87	474	480	476.2	8.92	−1	−1	
P1000-15000	582	589	586.6	15.43	585	591	588.5	15.56	585	592	587.0	9.56	−3	−3	
P1000-20000	653	661	658.2	14.03	657	660	658.4	12.21	653	658	656.0	7.68	−4	0	
P1000-25000	702	708	704.8	18.54	704	710	707.4	17.95	703	708	705.2	13.68	−2	−1	
P1000-30000	741	747	744.6	19.95	744	750	746.9	23.98	741	746	743.5	19.97	−3	0	
#Better	–	–	–		0/20	3/20	3/20		0/20	6/20	4/20				
#Equal	–	–	–		10/20	8/20	1/20		11/20	10/20	1/20				
#Worse	–	–	–		10/20	9/20	16/20		9/20	4/20	15/20				
p-value	–	–	–		2.09E−3	1.93E−2	3.31E−3		4.04E−3	8.74E−1	5.85E−1				
Note: The bold numbers in the table highlight the dominating results between the compared algorithms in terms of Best, Worst and Avg values.

Even though both IDTS and IDTS1 obtain 10 equal results, the former can achieve 10 better results (against 0 for IDTS1). The small p-values (<0.05) in terms of Best and Avg confirm that the reported differences between IDTS and IDTS1 were statistically significant. This experiment proves that the perturbation strategy adopted is an important way of diversification that makes the algorithm able to better explore the search space. Both IDTS and IDTS2 obtain 11 equal results while the former achieves nine better results than the latter. The small p-value (<0.05) indicates that IDTS is better than IDTS2. The above indicates that adopting DROP and INSERT operations in the perturbation procedure can enable the algorithm to reach a better performance.

Conclusions

An efficient stochastic local search algorithm IDTS was proposed to find the minimum set of feedback vertices in graphs. It begins with a low-complexity greedy initialization procedure, and alternates between a thresholding search stage and a descent stage. The IDTS algorithm has two innovative components, the solution-accepting strategy used in the thresholding search stage and the frequency-guided strategy in its perturbation procedure. The thresholding search stage involves an adjustable thresholding parameter δ that controls the search behavior and algorithm performance. Since fine-adjusting this parameter for a given problem instance can bring better solutions, it will be meaningful to study self-adaptive mechanisms to automatically adjust this parameter during the search.

Experimental evaluations on 101 diverse graphs proved the dominance of IDTS over the state-of-the-art SA (Galinier, Lemamou & Bouzidi, 2013) and BPD (Zhou, 2016) algorithms. Particularly, it discovered 24 new best-known results (improved upper bounds), and reached the best-known or known optimal results of 75 other graphs. We also applied our algorithm to the case of undirected graph of the problem and showed its competitiveness against the SALS algorithm (Qin & Zhou, 2014). Besides, we conducted experiments to understand how each ingredient of IDTS (the thresholding and the short term learning-based perturbation) contributes to the algorithm performance.

Finally, it will be of interest to study the proposed framework for other critical vertex problems, such as the critical node detection (Béczi & Gaskó, 2021) and finding the nodes with the highest betweenness-centrality scores (Mirakyan, 2021).

Supplemental Information

Supplemental Information 1 101 benchmark instances used in this article.

These 101 benchmark instances are classified into five categories. No optimal solutions are known for the instances of the first to forth categories, while optimal solutions are known for the instances of the fifth category.

Click here for additional data file.

Supplemental Information 2 The undirected instances with ER* in their names.

These instances have ER* in their names, where the number of vertices is 100000, and the number of edges is in the range of [100000, 1000000].

Click here for additional data file.

Supplemental Information 3 The undirected instances with RR* in their names.

These instances have RR* in their names, where the number of vertices is 100000, and the number of edges is in the range of [100000, 1000000].

Click here for additional data file.

Supplemental Information 4 The best results obtained by our algorithm for each instance.

Each data indicates a series of the best objective values, the ever-found optimal solutions, and the corresponding computation time obtained by our algorithm in the search process.

Click here for additional data file.

Supplemental Information 5 Our code for solving the feedback vertex set problem (Directed version).

Click here for additional data file.

Supplemental Information 6 Our code for solving the feedback vertex set problem (Undirected version).

Click here for additional data file.

We are grateful to the Academic Editor and anonymous reviewers for their valuable suggestions and comments, which helped us to improve the article.

Additional Information and Declarations

Competing Interests

Author Contributions

Data Availability

Jin-Kao Hao is an Academic Editor for PeerJ.

Wen Sun conceived and designed the experiments, performed the experiments, analyzed the data, prepared figures and/or tables, authored or reviewed drafts of the article, and approved the final draft.

Jin-Kao Hao conceived and designed the experiments, analyzed the data, authored or reviewed drafts of the article, and approved the final draft.

Zihao Wu performed the experiments, analyzed the data, performed the computation work, prepared figures and/or tables, and approved the final draft.

Wenlong Li analyzed the data, performed the computation work, prepared figures and/or tables, authored or reviewed drafts of the article, and approved the final draft.

Qinghua Wu conceived and designed the experiments, authored or reviewed drafts of the article, and approved the final draft.

The following information was supplied regarding data availability:

The raw data is available in the Supplemental Files.

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
