# Peer review of "Dynamic thresholding search for the feedback vertex set problem"

_PeerJ Computer Science, doi:10.7717/peerj-cs.1245_

## Round 0.1 · original submission · Major Revisions

We have received two reviews for your submission "dynamic thresholding search for the feedback vertex set problem" which suggested that mayor revisions are in order. Most importantly, the two reviewers expect that you should tackle larger instances than those used in the paper. Also, they propose to introduce and test at least two new algorithms, namely the BPD algorithm and the SA by S. Qin. In addition, the algorithm and the paper seems not easy to follow.

Reviewer 1 ·

Basic reporting

This manuscript offered a new heuristic algorithm for solving the minimum feedback vertex set problem on directed graphs. The performance of this algorithm was compared with that of the simulated annealing algorithm on many relatively small graph instances. It was demonstrated that this new algorithm could achieve better solutions.

This algorithm has several intuitive considerations, each of which brings certain hyperparameters. The optimal values of this set of hyperparameters could be problem instance dependent. In comparison with simulated annealing, the number of hyperparameters is much larger.

The description of the algorithm is not easy to follow, but still comprehensible. It would be beneficial to readers to further improve the description.

A major weakness of this work is that the authors only tested small graphs (size not exceeding 1000 vertices). It is not yet convincing that it works equally good on larger graph instances containing say, of the order of one million vertices. The SA and the BPD message-passing algorithm as cited in the manuscript can efficiently on such large graph instances. Maybe it is a good extension to apply the algorithm on random directed graphs (of in-degree and out-degree say 10), and how the computing time and the relative FVS size scale with graph size N.

Experimental design

As I suggested above, it is desirable to apply the code on large graph instances, and compare the results with results obtained by SA or by message passing BPD.

Validity of the findings

As far as I can see, the results are valid.

Additional comments

How about the applicability of the algorithm to undirected graphs? Highly likely that it will also beat the SA algorithm adapted by S. Qin (EPJB 2016)?

Reviewer 2 ·

Basic reporting

In this manuscript, the authors introduced the dynamic thresholding search for the feedback vertex set problem. The work is interesting and the results seem correct. The following issues should be consider before the manuscript can be accepted.

1. The shorthand cannot appear alone when it is used for the first time. For example, the FVSP, VLSI in the Abstract.

2. The definitions proposed in section 1.1 are not clear and hard to understand

3. The related results referenced in section 1 could be first introduced as Preliminaries in detail in a separate section.

4. Except the SA algorithm, the authors should consider other kinds of algorithms, such as the BPD algorithm, to be the references to assess the IDTS.

5. In addition, the real-world networks and the artificially generated scale-free network should be used, not just the 40 benchmark instances generated in 1998.

Experimental design

no comment

Validity of the findings

no comment

Additional comments

no comment

---

## Round 0.2 · Minor Revisions

The paper was found to be improved and it is now accepted. However, I have only a few editorial revisions. Essentially, these concern the tables and some of the figures, which should be made of the same letter size. I would take the table 2 as the default size and every table should be brought to the same default size. This concerns table 1, 3, 4, 5, 6, 7 (probably 8), and 9.
Also the figures have to be brought to a standard letter size. This concerns mainly Figure 10 and 11, where I don't see anymore in standard resolution what is in the figures. Maybe you have to do also a part of the other figures in the paper.

Reviewer 2 ·

Basic reporting

no comment

Experimental design

no comment

Validity of the findings

no comment

Additional comments

All my concerns have been addressed.

---

## Round 0.3 · accepted · Accept

This is the acceptance of the manuscript. Congratulations!